# Short Communication: Understanding the Barriers to Cervical Cancer Prevention and HPV Vaccination in Saudi Arabia

**DOI:** 10.3390/v16060974

**Published:** 2024-06-18

**Authors:** Jobran M. Moshi, Aarman Sohaili, Hassan N. Moafa, Ahlam Mohammed S. Hakami, Mohsen M. Mashi, Pierre P. M. Thomas

**Affiliations:** 1Department of Medical Laboratory Technology, Faculty of Applied Medical Sciences, Jazan University, P.O. Box 114, Jazan 45142, Saudi Arabia; jmoshi@jazanu.edu.sa; 2Institute of Public Health Genomics, Genetics and Cell Biology Cluster, GROW Research School for Oncology and Development Biology, Maastricht University, 6229 ER Maastricht, The Netherlands; aarman.sohaili@outlook.com; 3Department of Epidemiology, College of Public Health and Tropical Medicine, Jazan University, P.O. Box 114, Jazan 45142, Saudi Arabia; moafa@jazanu.edu.sa; 4Department Obstetrics and Gynecology, Jazan University, P.O. Box 114, Jazan 45142, Saudi Arabia; ahlamhakami@jazanu.edu.sa; 5Department of Microbiology, Jazan Armed Forces Hospital, Jazan 1568422, Saudi Arabia; mohsen-mashi@hotmail.com

**Keywords:** HPV, cervical cancer, HPV vaccine, prevention, Saudi Arabia

## Abstract

Cervical cancer, along with other sexual and reproductive health and rights (SRHR) conditions, poses a significant burden in the Kingdom of Saudi Arabia (KSA). Despite the availability of effective preventive methods such as vaccinations, particularly against the Human Papillomavirus (HPV), awareness about such preventive methods and HPV vaccination remains alarmingly low in the KSA, even with governmental effort and support. While many women are aware of the risks, the uptake of the HPV vaccine remains below 10% (7.6%) at the country level. This highlights the urgent need for Knowledge, Attitude, and Practice (KAP) at the community level to raise awareness, dispel misconceptions, and empower women to embrace vaccinations. Additionally, there is a need to revitalize the cancer registry system to better track and monitor cervical cancer cases. This short communication aims to map these barriers while identifying opportunities for impactful research. Drawing from the scientific literature, government reports, and expert insights, we highlight the challenges surrounding the tackling of HPV. By exploring diverse sources of knowledge, this paper not only highlights current obstacles but also proposes actionable solutions for future interventions.

## 1. Introduction

Cancers and neoplasia stand out as major contributors to the global burden of mortality and morbidity. Cervical cancer is the fourth most common cancer in women worldwide, with an estimated 660,000 new cases and 350,000 attributable deaths annually as of 2022 [1]. Furthermore, cervical cancer disproportionately impacts the Global South, necessitating urgent public health action [2].

A significant portion of the burden of cervical cancer is preventable, given that one of the major contributing factors is the presence of carcinogenic strains of the Human Papilloma Virus [HPV]. Particular HPV strains, especially subtypes 16 and 18, have, in fact, been documented to be involved in the development of precancerous cervical lesions and neoplasia. The implementation of preventive measures, notably HPV vaccination, could potentially avert up to 70% of these cases [3].

Countries located in the Middle East are explicitly affected by the burden of cancers, which holds true for Saudi Arabia. The KSA is home to 37 million inhabitants [4]. Alongside other nations in the Middle Eastern and Northern African Region (MENA), the KSA is experiencing a shift in the mortality and morbidity patterns [5]. The Saudi healthcare system is facing increased strain due to the rising prominence of cancer and other non-communicable diseases.

While the burden of mortality and morbidity from cervical cancer is low in the KSA when compared with other regions in the world, it is swiftly evolving into a matter of significant public health concern [6]. Numerous barriers impede prevention, screening and treatment of cervical cancer, leading to a complex challenge for the healthcare system [7]. There exists multiple knowledge and research gaps, particularly in the realm of preventive efforts such as HPV vaccination [8].

Recognizing the preventable morbidities and mortalities associated with cervical cancer, as well as the global efforts and resources dedicated to combatting HPV, it is crucial to address the current barriers to cervical cancer and HPV vaccination efforts in the KSA. The aim of this paper is to map the current barriers to cervical cancer and HPV vaccination efforts in KSA, and to outline opportunities for impactful research. This paper aims to bring findings from the scientific literature, government reports, and the experience of experts on the ground to the forefront in order to shed light on the challenges and outline solutions for the future. Additionally, this study aims to provide a comprehensive overview of the existing knowledge gaps and methodological limitations surrounding HPV and cervical cancer, thereby directing future research efforts towards generating the data needed to inform more effective public health strategies.

By enriching the literature with comprehensive data on these aspects, this study seeks to provide valuable insights that can inform the development of a detailed national strategy to eliminate cervical cancer in the KSA.

## 2. Materials and Methods

The study utilized a dual approach to examine the burden of cervical cancer and vaccination efforts in the KSA. The first part involved a mapping review that integrated both qualitative and quantitative data to chart the KSA’s progress in tackling cervical cancer and in their vaccination efforts.

To guide and contextualize the research, a mapping review framework established by Paré and Kitsiou [9] was used to identify extensive literature across multiple electronic databases. To ensure broad coverage and minimize potential gaps, the following combination of databases was used: (“Human Papillomavirus” [MeSH] OR HPV [Title/Abstract]) AND (“Cervical Neoplasms” [MeSH] OR “Cervical Cancer” [Title/Abstract]) AND (“Papillomavirus Vaccines” [MeSH] OR “HPV Vaccine” [Title/Abstract]) AND “Saudi Arabia” [MeSH]. The search was aimed at identifying relevant articles, with an emphasis on randomized controlled trials, systematic reviews, and reputable grey literature from sources like the WHO and the Saudi Ministry of Health.

Following the literature search, a meticulous screening process was employed to manage the references efficiently, eliminate duplicates, and select studies that aligned with the research question. Articles selected for a full-text review were individually assessed by the authors to determine their alignment with the inclusion criteria. This was supplemented by ‘reverse snowballing’ to capture any additional relevant studies.

The inclusion criteria for this review were defined as follows:(1) The types of articles considered included empirical research studies conducted in Saudi Arabia.(2) The review was limited to articles published from 2015 onward to ensure the relevance of the data in reflecting current trends and developments in HPV trends.(3) The focus of the review was on studies that analyzed specific variables related to HPV in the KSA.(4) Only articles originally written in English and Arabic were included, to accommodate the linguistic scope of the study.

The selected studies were then systematically charted and recorded, detailing the citation, location, method of evaluation, and key HPV indicators assessed.

As a complement to the analysis, expert opinions were gathered from notable Saudi Arabian experts across different fields. These experts were chosen for their extensive experience in addressing HPV from the perspective of various disciplines (epidemiology, microbiology, and public health). Their broad expertise provided unique and valuable perspectives on the challenges and strategies related to cervical cancer and HPV vaccination. Written informed consent was secured from the experts before the beginning of the study.

## 3. Results

### 3.1. Awareness

Sexual and Reproductive Health and Rights [SRHR] awareness is consistently low in the MENA region, including the KSA, with a particular deficiency in understanding cervical cancer among women, as noted in [10]. The overall awareness levels remain largely undocumented at the population level [10]. Notably, prior to 2022, SRHR topics, including cervical cancer, were absent from the school curricula in Saudi Arabia. Subsequent updates introduced a modest inclusion of these subjects in high school, primarily within the health fields at universities [11].

Despite these efforts, some awareness gaps persist, extending to medical professionals. Recent data highlight a significant lack of information and referrals for cervical cancer screening among women. Although the Ministry of Health aims for implementation, there remain challenges in realizing these intentions.

Questions arise about the integration of cervical cancer prevention, such as HPV vaccination, into the healthcare system. A recent nationwide study in the KSA focused on sexually active Saudi women aged 21 to 65, revealing concerning trends in cervical cancer screening and HPV vaccine uptake [8]. The findings showed that only 22.1% of participants had undergone cervical cancer screening, with merely 7.6% receiving the HPV vaccine. Despite some positive attitudes towards screening and vaccination, a significant 84.1% of participants lacked the essential knowledge about cervical cancer screening.

These observed gaps in the uptake of preventive measures are associated with sub-optimal communication between primary care physicians and the target population, insufficient educational campaigns, and a lack of health promotion programs. This trend continues to persist, with 40% of those declining the HPV vaccine citing reasons like limited awareness and concerns about injection-related adverse events [10].

Insufficient awareness, even among healthcare professionals, leaves women without the essential knowledge about cervical cancer and screening. Addressing these challenges is vital for enhancing awareness of SRHR and promoting preventive measures in the region. Furthermore, there is a public inquiry regarding the incorporation of HPV vaccination into the Saudi healthcare system, ensuring its availability for young women at general practices without charge.

### 3.2. Vaccine Hesitancy

The prevention of cervical cancer faces additional challenges beyond awareness, with vaccine hesitancy standing out as a significant hurdle [10]. In 2020, a study uncovered that only 2% of Saudi females had received the HPV vaccine [12]. Surprisingly, even among populations with a solid comprehension of cervical cancer risks, the vaccine uptake remains low, with less than 10% of females having received any form of HPV vaccination [13]. Surprisingly, this hesitancy extends to healthcare students, including those in medical, nursing, and dentistry fields. Cultural and religious factors also appear to play a role in shaping attitudes, with the factor of traditional medicine holding importance within communities [14].

Informal networks, such as family and friends, seem to play a vital role in disseminating information and encouraging vaccination [15]. Leveraging these networks could contribute to overcoming the hesitancy surrounding vaccines, especially in communities where cultural and religious factors heavily influence decision-making.

It is worth noting that the KSA approved the commercial vaccines Gardasil and Cervarix in 2010, with widespread implementation occurring in 2017. The Ministry of Health then began integration of the vaccines into the national immunization schedule, making them available to girls aged 9–13 [16]. However, the implementation seems to currently be in the pilot phase, limited to selected regions of the KSA, and lacks a comprehensive overview.

### 3.3. Treatment

Cancer is increasingly recognized as a major public health concern and economic burden in the KSA. The Saudi government has prioritized the healthcare system, investing in infrastructure, and providing education and training opportunities for health professionals. Currently, there are 15 specialized oncology care settings situated in major cities, catering to around 80% of the Saudi population, with ongoing efforts to extend such services to less populated areas [17,18].

Cancer care in the KSA is accessible to all Saudi citizens free of charge, and non-citizens can avail themselves of these services through health insurance and non-profit organizations. The current cancer care model in the country follows a “find it and fix it” approach, lacking comprehensive risk factor surveillance programs and proactive interventions [19]. Patients usually access cancer care settings through various routes, including primary or secondary healthcare referrals, screening centers diagnosis, or emergency room presentations [20].

Aligned with the Saudi Vision 2030, a new care model is being developed, focusing on the prevention of non-communicable diseases [NCDs] and involving the private sector more actively in cancer care [19]. Despite this, several challenges to cervical cancer exist in the KSA. Improving the quality of cancer care is a continuous process. One area of improvement would be the strengthening of current cancer registries. The Saudi health authority would, in fact, benefit from more capability to track and monitor current cancer trends, care outcomes, and survival rates [21]. This is especially true for the cases where Saudi citizens seek healthcare outside of the country. A rigorous registry would, in fact, help tailor current outreach efforts and understand current needs.

## 4. Way Ahead and Conclusions

In light of the current barriers to cervical cancer prevention and HPV vaccination in the KSA, a multidimensional approach ought to be outlined. A pervasive challenge is the lack of awareness, with certain segments of the Saudi population vaguely acknowledging the risk of cervical cancer, while overall community-level awareness remains low. Additionally, awareness of HPV vaccination options is consistently limited, emphasizing the need for innovative strategies to educate adolescents and the individuals of reproductive age.

To enhance awareness, health authorities should delve into current knowledge gaps, identify contentious ideas, and understand the influence of traditional and religious beliefs. Recent data indicate suboptimal awareness among healthcare professionals, highlighting the importance of targeted efforts to inform medical practitioners, nurses, and healthcare providers. This, in turn, can create a ripple effect, empowering patients to make informed decisions about their sexual and reproductive health.

A holistic approach to raising awareness, coupled with an understanding of community-level health-seeking behavior, presents a significant intervention opportunity. Furthermore, existing shortcomings in cervical cancer screening in the KSA, exacerbated by underutilized screening opportunities and insufficient referrals from primary care practitioners, require attention. A thorough examination of the current care pathway is essential to structuring strategies for optimal outcomes.

Augmenting screening efforts could involve incorporating HPV testing, potentially as part of pre-marital screenings for sexually transmitted infections. This approach aligns with a broader strategy to improve reproductive health outcomes. The healthcare sector’s deficiencies extend to the cancer registration infrastructure, where gaps in gathering accurate and up-to-date cancer epidemiology data at the national level persist. Establishing a well-functioning database for cervical cancer cases in the KSA is crucial for tracking trends, identifying high-risk groups, and developing evidence-based prevention strategies. A robust information system would contribute significantly to improving the overall effectiveness of cervical cancer prevention initiatives in the KSA.

## Data Availability

No new data were created for this paper; the literature and documents used can be made available upon reasonable request.

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
