# Peer review of "Short Communication: Understanding the Barriers to Cervical Cancer Prevention and HPV Vaccination in Saudi Arabia"

_viruses, 2024, doi:10.3390/v16060974_

Round 1
Reviewer 1 Report
Comments and Suggestions for Authors
1. In the abstract in the sentences ‘Despite the availability of 20 effective prevention methods such as vaccination, particularly against Human Papillomavirus 21 (HPV), awareness about HPV vaccination remains alarmingly low in the country. Despite govern- 22 ment efforts and support, the awareness of HPV vaccination remains low in the KSA.’ A lack of awareness of vaccination is repeated 2 times. Paraphrase the sentences and eliminate repetitions
2. Why use case-control studies if the data from these studies are not provided? Only general information is prescribed. I suggest adding the data already available from the articles on the association of cervical cancer with HPV in KSA
3. Why were articles from 2001 used as inclusion criteria? Although the reference list is absolutely up-to-date. Revise and correct.
4. In my opinion, it seems incorrect to include a survey of experts in the field. Taking into account that the authors of the article did not even specify who these experts are and what they said. At the same time, if the survey is done in the format of an interview, it will no longer be a scientific article. I suggest removing this paragraph from the materials and methods or rewriting it in a more logical way. For example, it would be possible to cite the research of these experts as an example of their opinion on this topic
Author Response
Dear Reviewer,
We greatly appreciate the time that you took to review our manuscript. On behalf of the research team, I have listed answers to your points below:
- Sentences and repetitions in the abstract have been removed and redundant phrases removed
- Thanks for this point, we have updated the methodologies and types of articles that are considered in this manuscript. The previous formulation did point at a holistic approach while this paper's aim remains to raise awareness and contribute to the dicsussion rather than providing a systematic overview of the literature. The methodology section was streamlined accordingly.
- Thanks for your valuable comment, we have updated the inclusion timeframe to 2015 to better represent the recent trends
- We have updated the description of the role of the experts in the present study. We have now specified that the opinions of the different experts was considered. The section in the methodology wads updated to better represent the contributions. We furthermore went on to add some new references that were shared by the experts in the main body of the text. Most the newly added sources come from governmental sources, and help paint a more descriptive picture of the current situation on the ground in the KSA
We would like to once again thank you for helping us refine this paper. We have furthermore undertaken a thorough review of the paper for english and Grammar.
The research team and myself would like to state that the present paper intends to pave the way for future scientific research on the topic in the KSA.
Reviewer 2 Report
Comments and Suggestions for Authors
This manuscript (short communication) provides overview of the existing knowledge gaps (awareness, vaccine hesitancy, treatment) and methodological limitations surrounding HPV and cervical cancer in the Kingdom of Saudi Arabia. Based on the data, the authors have outlined the need for further actions and directions for improving the situation. This short communication could be interesting/important for medical doctors, health care providers and health care politicians in the Middle Eastern and also other countries.
Unfortunately, there are a lot of spelling mistakes in the manuscript that need to be corrected.
Comments:
Line 19 - “Cancer” must be written in lowercase;
Line 20 - “kingdom” must be capitalised;
Line 25 - “ata” must be deleted;
Line 36 - “Neoplasia” must be written in lowercase;
Lines 41, 53, 59, 68, 103, 110, 112, 116,119, 138 - “Cervical Cancer” must be written in lowercase;
Line 44 - “Precancerous” must be written in lowercase;
Line 48 - after "The Kingdom of Saudi Arabia" the abbreviation "KSA" must be inserted;
Lines 70, 173 - “Cervical” must be written in lowercase;
Line 100 - “extensive” instead of ex-tensive”;
Lines 101-102 – “(Epidemiology, Microbiology and Public Health)” must be written in lowercase;
Line 163 - (17, 18) instead of (17)(18);
Line 182 - “KSA” instead of “the Kingdom of Saudi Arabia”;
Line 193 – “Sexual and Reproductive Health” must be written in lowercase;
Lines 219-260 - References are not formatted according to journal requirements - Journal references must cite the full title of the paper, page range or article number, and digital object identifier (DOI) where available.
References 8, 18, 20 - article number is missing;
Despite being available, no DOI is provided for any of the cited journal articles.
Comments on the Quality of English Language
English as such is fine, but there are a lot of spelling mistakes, especially in names where lowercase letters should be used.
Author Response
Dear Reviewer 2,
Many thanks for taking the time to review our manuscirpt, we appreciate your time. On behalf of the research team and myself, I wanted to provide you with an answer to your points below
- We went ahead and corrected all of the spelling mistakes that were brought to our attention. We furthermore had a native english speaker go over the manuscript once again.
- We have updated the references and references style to match the journal requirements. It now features the DOI for every article.
- For references 8, 18 and 21, it appears that they are all from the Curerus platfom, that does not seem to provide article numbers. Kindly advise on how to proceed from here
Thanks again for your valuable input.
Round 2
Reviewer 1 Report
Comments and Suggestions for Authors
The author of the paper has taken into account all the comments and corrected the paper